# COVID-19 Therapeutic Potential of Natural Products

**DOI:** 10.3390/ijms24119589

**Published:** 2023-05-31

**Authors:** Zhaoxuan Low, Rafidah Lani, Vunjia Tiong, Chitlaa Poh, Sazaly AbuBakar, Pouya Hassandarvish

**Affiliations:** 1Tropical Infectious Diseases Research & Education Centre (TIDREC), Universiti Malaya, Kuala Lumpur 50603, Malaysia; lowzx95@gmail.com (Z.L.); sazaly@um.edu.my (S.A.); 2Department of Medical Microbiology, Faculty of Medicine, Universiti Malaya, Kuala Lumpur 50603, Malaysia; rafidahl@um.edu.my; 3Centre for Virus and Vaccine Research, School of Medical and Life Sciences, Sunway University, Petaling Jaya 47500, Malaysia; pohcl@sunway.edu.my

**Keywords:** COVID-19, SARS-CoV-2, antiviral, therapeutics, natural products, traditional Chinese medicines, plant extracts

## Abstract

Despite the fact that coronavirus disease 2019 (COVID-19) treatment and management are now considerably regulated, severe acute respiratory syndrome coronavirus 2 (SARS-CoV-2) is still one of the leading causes of death in 2022. The availability of COVID-19 vaccines, FDA-approved antivirals, and monoclonal antibodies in low-income countries still poses an issue to be addressed. Natural products, particularly traditional Chinese medicines (TCMs) and medicinal plant extracts (or their active component), have challenged the dominance of drug repurposing and synthetic compound libraries in COVID-19 therapeutics. Their abundant resources and excellent antiviral performance make natural products a relatively cheap and readily available alternative for COVID-19 therapeutics. Here, we deliberately review the anti-SARS-CoV-2 mechanisms of the natural products, their potency (pharmacological profiles), and application strategies for COVID-19 intervention. In light of their advantages, this review is intended to acknowledge the potential of natural products as COVID-19 therapeutic candidates.

## 1. Introduction

The coronavirus disease 2019 (COVID-19) pandemic represented an unprecedented disaster to the health and welfare of humanity. Since May 2023, the number of cases and deaths worldwide resulting from COVID-19 has increased to an unanticipated number of ≈689 million and 6.88 million, respectively [1]. As a countermeasure to the continued worsening of the COVID-19 pandemic, enormous research efforts have been dedicated to vaccine development, which led to the rapid production of several notable vaccines, such as BNT162b2, AZD1222, mRNA-1273, and CoronaVac [2]. However, major obstacles to achieving herd immunity for COVID-19 include inequity in vaccine distributions, vaccine non-responders, the emergence of immune-escape variants of SARS-CoV-2, and vaccine hesitancy among the global population. Such skepticism revolving around vaccines was not completely unfounded, as multiple life-threatening side effects, including thrombosis [3], myocarditis [4], pericarditis [5], and even death, have been reportedly associated with vaccine use. With the uncertain long-term efficacy of currently available vaccines, and alarming number of infections and deaths caused by the pandemic, the Food and Drug Administration (FDA) issued an emergency use authorization (EUA) for several monoclonal antibody therapies, such as sotrovimab, tocilizumab, casirivimab, imdevimab, and baricitinib for treating or preventing COVID-19 infection [6,7]. Additionally, remdesivir represents the first direct-acting antiviral drug fully approved by the FDA for treating COVID-19. Ongoing efforts are being made through continuous collaborations between the FDA and pharmaceutical companies, researchers, and manufacturers to accelerate the development and discovery of new therapeutic drugs to prevent and treat COVID-19. Thus far, the standard treatment options for hospitalized patients with COVID-19 include the use of antivirals (remdesivir), corticosteroids (dexamethasone), monoclonal antibodies (baricitinib and tocilizumab), and oxygen supplementation care [8].

Unfortunately, most of these interventions are predominantly available to nations that possess the financial means to secure early access. This is largely due to patent holders resisting the call for licensing agreements that would enable generic manufacturers to produce similar antivirals or vaccines [9]. Consequently, low-income countries, which often have weaker healthcare systems, higher poverty rates, and limited access to vital resources such as medical supplies and vaccines, have been disproportionately affected [10]. Additionally, many low-income countries heavily rely on tourism and exports, both of which have been severely disrupted by the pandemic [11]. Hence, these countries have experienced a significant decline in revenue, making it more challenging to invest in crucial services such as healthcare and education, thus exacerbating existing inequalities.

As a result, effective, conveniently accessible, and economical antiviral targeting of SARS-CoV-2 is mandatory, especially for low-income countries. Nature is an excellent reservoir of effective compounds that can be directly used as pharmaceuticals or further developed into new drugs. Natural compounds derived from plants have been used as medicine throughout history and are considered safe compared to synthetic products, which are costly to be synthesized. Thus, a rigorous evaluation of the potential natural compounds, involving controlled clinical trials in conjunction with agricultural advancement to upscale the production of effective natural antivirals, is needed. In this way, this might lead to the availability of anti-SARS-CoV-2 drugs at a reasonable cost while serving as a source of income in low-income countries. The availability of natural antivirals might play an essential role in supporting the response to COVID-19 in low-income countries, helping to make healthcare more affordable, accessible, and sustainable. 

This review aims to describe natural products as potential antivirals against SARS-CoV-2 by emphasizing their efficacies, mode of action, and therapeutic potentials as novel therapies against COVID-19. This review concentrates explicitly on the possible natural products that have substantiated proof of concept against SARS-CoV-2, excluding other viruses within the Coronaviridae family. The selection criteria prioritize compounds with promising outcomes that warrant consideration for clinical trials. Additionally, the review incorporates findings from recent randomized clinical trials (RCTs) and offers insights into future perspectives for enhancing the draggability of these natural compounds. 

## 2. SARS-CoV-2 and Its Life Cycle

COVID-19 is caused by SARS-CoV-2, which is an enveloped, positive-sense, single-stranded RNA virus. The SARS-CoV-2 virion is a spherical particle with a diameter ranging from 60 to 140 nm [12]. It contains a nucleocapsid (N)-encapsulated genome which is enclosed in an envelope that is associated with three structural proteins: membrane (M) protein, S protein, and envelope (E) protein [13,14,15]. The whole-genome (~30 kb) alignment of SARS-CoV-2 with other available genomes of *betacoronaviruses* showed high sequence similarity to the SARS-related coronavirus RaTG13 (96%) found in the Yunan province of China, suggesting that SARS-CoV-2 is likely to evolve from the bat RaTG13 strain [16]. The positive-strand genome contains a 5′-cap and a 3′-poly-A tail with two-thirds of its genome comprising two large open reading frames (ORFs), ORF1a and ORF1b, at the 5′ end that encodes for sixteen non-structural proteins (nsp1-16) [14,17,18,19]. Mechanistically, a programmed −1 frameshift that occurs between ORF1a and ORF1b allows for the production of two polyproteins, polyprotein 1a (pp1a) and polyprotein 1ab (pp1ab), which are further proteolytically cleaved by nsp3 (papain-like protease, PL^pro^) and nsp5 (chymotrypsin-like protease, CL^pro^) into sixteen individual non-structural proteins (NSPs) [14,20,21]. In contrast, ORFs spanning the remaining one-third of the genome at the 3′ end encode accessory proteins and four structural proteins: S, E, M, and N proteins [14,22,23]. Deciphering the molecular mechanisms of the virus life cycle and a good understanding of the virus biology is necessary for the development of any novel, effective therapeutic strategies.

Broad tropism of SARS-CoV-2 infection has been shown in various tissues, including lung, small intestine, kidney, and pancreas [24,25,26,27]; however, most severe SARS-CoV-2 infections in patients are commonly associated with extensive lung damage that results in pneumonia, acute respiratory failure, and death due to abundant amounts of the SARS-CoV-2 principal receptor, angiotensin-converting enzyme 2 (ACE2) being expressed in the airway epithelium, and the respiratory tract being the first portal of entry [27,28,29,30]. SARS-CoV-2 virus attachment and entry into host cells are mediated by two major subunits of the S protein: S1 and S2 subunits [15]. The S1 subunit contains a receptor-binding domain (RBD) that recognizes and binds strongly to the ACE2 receptor on the host cell, enabling virus attachment to the host cell surface [15,31,32,33]. Prior to membrane fusion, the S protein is primed through proteolytic cleavage at two sites: furin (S1-S2) and transmembrane serine protease 2 (TMPRSS2; S2′) cleavage sites. Presumably, the cleavage of the S1-S2 site by furin promotes conformation changes that make the S2′ site accessible for TMPRS22 cleavage, triggering the exposure of a fusion peptide formed via the two-heptad repeats (HR1 and HR2) in the S2 subunit [32,33,34,35]. Alternatively, SARS-CoV-2 can also utilize cathepsin L to enter cells via the endosomal pathway, where membrane fusion could occur independently of TMPRSS2-mediated S2′ cleavage [36]. 

On entering the host cell, the viral genome is released into the cytoplasm where it is rapidly subjected to the translation by host cell ribosomes to produce pp1a and pp1ab. These polyproteins are further proteolytically processed into individual NSPs, forming the replication and transcription complex (RTC) [37]. The major components of the RTC comprise the NSP12 that functions as the core RNA-dependent RNA polymerase (RdRp) and its nsp7 and nsp8 cofactors [38]. Additionally, the nsp13 also constitutes the core RdRp complex through its interaction with nsp8 and functions as a helicase [39]. On the other hand, the nsp10-14-16 complex possesses the core enzymatic functions involved in RNA proofreading and RNA capping to ensure the synthesis of mature viral RNA [37,40,41,42]. Importantly, the replication and transcription processes are carried out in virus-induced organelles, such as double-membrane vesicles (DMVs), convoluted membranes (CMs), and vesicle packets (VPs) [43]. These specialized membranous structures are likely formed by nsp3, nsp4, and nsp6 [43,44,45], which help compartmentalize newly synthesized RNA genomes and transcripts to protect them against host cell defenses and subsequent degradation [46,47]. 

Newly synthesized viral RNA genomes in the specialized membranous structures are transported through double-membrane-spanning pore complexes where viral assembly and budding take place in an endoplasmic-reticulum-Golgi intermediate compartment (ERGIC) in a single-membrane vesicle (SMV) lined by viral S, E, and M glycoproteins [48]. The primary function of the N protein is to protect and encapsidate the viral RNA genome by forming the ribonucleoprotein (RNP) complex that is ultimately packaged into the virion via association with the M protein during virion assembly [49,50,51,52]. Hence, it is conceivable that the interactions of N proteins of the RNP complexes with the M proteins drive the assembly and budding process of virus particles by stabilizing the RNP complex, while E protein lining the SMV also plays an indispensable role in completing these processes. Lastly, the fusion of the mature virion-containing vesicles with host cell membranes promotes the egress of SARS-CoV-2 virions via exocytosis [48] (Figure 1).

## 3. Methodology

The literature review search was conducted using pertinent keywords such as COVID-19, SARS-CoV-2, antiviral therapeutics, natural products, traditional Chinese medicines (TCM), plant extracts, and pure natural compounds. Boolean operators (AND, OR, NOT) were employed to refine the search, and scholarly databases, including Google Scholar, PubMed, and Scopus, were utilized for the preliminary search. Following this, natural products that demonstrated in vitro or in vivo activity against SARS-CoV-2 were included, while those solely reported in silico were excluded. The search findings were organized in Table 1 categorizing potential natural compounds into traditional Chinese medicines (TCM), natural extracts, and pure natural compounds. Table 1 also summarized essential details such as the structure, potency (IC_50_), toxicity (CC_50_), and mechanism of action of these natural compounds. To visually represent the inhibitory mechanisms of the identified natural products, Figure 2 was created, illustrating the distinct stages of the SARS-CoV-2 life cycle targeted by these inhibitors. Finally, the literature search scope was expanded to generate results that might guide future application strategies for natural products against SARS-CoV-2. Finally, a search was conducted on the ClinicalTrials.gov website (https://clinicaltrials.gov/ accessed on 22 May 2023) to identify any existing RTCs about the aforementioned potential natural compounds.

## 4. Pre-Infection Inhibitors, Their Proposed Mechanisms, and Their Potency against SARS-CoV-2

Pre-infection inhibitors primarily act against viruses before the internalization phase, such as by blocking the viral attachment and entry into host cells. Moreover, antivirals that target the pre-infection stage might alter the cellular endocytic mechanisms, change the environment conditions (pH) or interact with the cellular receptor and proteolytic enzymes to prevent internalization and replication of the virus [89,90]. Pre-infection inhibitors are usually used as prophylactics to prevent the development of severe viral disease, reduce the risk of disease transmission, or modulate the immune system before exposure to pathogens.

Epigallocatechin gallate (EGCG) found in tea extracts was reported to bind to the RBD of extracellular SARS-CoV-2 directly, thus preventing the interaction of the virus with host receptor ACE2 and subsequent viral entry [91]. In further corroboration of the findings above, EGCG in green tea displayed a dose-dependent anti-entry effect against pseudotyped lentiviral vectors carrying SARS-CoV-2 S protein with an IC_50_ value of 2.47 μg/mL. In live viruses, EGCG could inhibit the viral plaque formation of SARS-CoV-2, MERS-CoV, and SARS-CoV. Using the SARS-CoV-2 vitronectin ELISA kit, researchers further reported that EGCG could simultaneously bind to the RBD of SARS-CoV-2 and compete with ACE2 binding, efficiently blocking the recruitment of these viruses to host cell surfaces [71]. A Phase 2 randomized clinical trial (NCT04446065) is currently ongoing to evaluate the prophylactic effect of Previfenon, a drug consisting of EGCG, in high-risk healthcare workers. The primary and secondary endpoints are the event of COVID-19 cases and the rate of positive cases for IgM and IgG anti-SARS-CoV-2. 

When highly sulfated polysaccharides were screened using surface plasmon resonance (SPR), RPI-27, RPI-28, fucoidans, and heparin could bind to the S-protein of SARS-CoV-2 [82]. Among all the polysaccharides tested, RPI-27, a high-molecular-weight branched polysaccharide, was identified as the most potent antiviral against SARS-CoV-2 with IC_50_ of 83 nM, more effective than remdesivir [82]. Other sulfated polysaccharides such as heparin, TriS-heparin, and non-anticoagulant low-molecular-weight heparin (NACH) also showed promising antiviral activity against SARS-CoV-2. An in silico study demonstrated that heparin could bind to the S-protein RBD (PDB ID: 6VW1), suggesting that sulfated polysaccharides might target the entry stage of SARS-CoV-2 [82]. Another study also reported that marine sulfated polysaccharides extracted from *Stichopus japonicus*, brown and red algae, showed dose-dependent antiviral activities against SARS-CoV-2. Among these polysaccharides, sea cucumber sulfated polysaccharide (SCSP) extracted from *Stichopus japonicus* was identified as the most potent polysaccharide with an IC_50_ of 9.10 μg/mL. SCSP could bind to the S protein carried by the pseudotyped virus and prevent its entry into the host cells [84].

Various bioactive compounds have been identified in *Vitis Vvinifera* leaf extract, and most of them were derivatives of quercetin, luteolin, kaempferol, apigenin, isorhamnetin, myricetin, chrysoeriol, biochanin, isookanin, and scutellarein [65]. *Vitis Vinifera* leaf extract was reported to have a virucidal effect against enveloped DNA viruses, HSV-1, and RNA viruses, such as SARS-CoV-2. Cotreatment of the virus and this extract to the Vero cells showed potent viral inhibition, suggesting that it might block virus attachment and thus prevent entry [65]. Temperature shift assays further confirmed this projection, and gene expression study demonstrated that *Vitis Vinifera* leaf extract strongly reduced the S protein expression in SARS-CoV-2-infected cells [65].

HIDROX, containing 40% of hydroxytyrosol (HT), is a commercial product produced by the olive oil industry and is used as a natural product to improve health. The anti-SARS-CoV-2 activity of HIDROX and HT was reported to be virucidal against the extracellular virus in a time and dose-dependent manner [69]. Additionally, by comparing the efficiency of their virucidal effects, HIDROX was found to be more potent than pure HT. In addition, both HICROX and HT were suggested to alter the structure of the SARS-CoV-2 S protein, which was reflected in the molecular weight increase in S1, S2, and RBD subunits when evaluated in western blotting. This may cause the S proteins to aggregate, leading to the virucidal effects observed in cells. On the other hand, HIDROX-containing cream also had virucidal activity against SARS-CoV-2, indicating that HT-rich olive extract could be applied to the skin for protective purposes to reduce transmission of COVID-19 [69]. 

Water extract of *Prunella vilgaris* (NhPV) was reported to inhibit the SARS-CoV-2 S protein-mediated virus entry [75]. It was observed that NhPV could hinder the entry of SARS-CoV-2 pseudotyped HIV that carries full-length S protein (IC_50_ = 30 μg/mL) or mutated S protein, D614, that enhanced virus infectivity. Time-of-drug addition assays revealed that NhPV treatment worked best when cells were pre-treated with NhPV 1 h before viral infection. ELISA assay further confirms that NhPV could bind to ACE2 and block the interaction of viral S protein to the ACE2 receptor, thus preventing viral entry. A combination of NhPV and anti-SARS-CoV-2 neutralizing antibody (SAD-S35) enhanced the blocking effect, so similar efficiencies could be achieved by reducing the compound concentrations. Moreover, 50 μg/mL of NhPV could prevent the CPE induced by wild-type SARS-CoV-2. By titrating the production of viral progeny, researchers also reported that 100 μg/mL of NhPV could completely prevent virus infection [75]. 

The mixture of *Agrimonia Pilosa* (AP) and *Galla rhois* (RG) in a ratio of 6:4 (APRG64) was reported to inhibit the formation of SARS-CoV-2 plaques by blocking SARS-CoV-2 entry into host cells, and their antiviral effects were comparable to remdesivir and chloroquine phosphate [70]. Ursolic acid, quercetin, and 1,2,3,4,6-penta-O-galloyl-β-d-glucose were identified as active compounds responsible for the anti-SARS-CoV-2 activity of APRG64. They significantly reduced the S protein in cell supernatants and could inhibit SARS-CoV-2 propagation. A molecular docking study showed that ursolic acid exhibited the highest binding affinity towards SARS-CoV-2 S RBD and the B.1.1.7 lineage S RBD with a binding energy of −9.5 kcal/mol and −9.0 kcal/mol, respectively [70]. 

Echinaforce, an alcoholic extract of *Echinacea purpurea* plants, has been previously evaluated for its benefit in treating the common cold and other respiratory tract infections [92,93,94,95]. Echinaforce was observed to have irreversible virucidal activity against HCoV-229E in viral infected pseudostratified respiratory epithelial cell culture, which better mimics the in vivo lung model. Moreover, Echinaforce also exhibited virucidal activity against MERS-CoV, SARS-CoV, and SARS-CoV-2, with complete inactivation after 50 μg/mL Echinaforce treatment [74]. The antiviral effect of Echinacea in clinical trial (NCT05002179) against respiratory viruses, including coronavirus (229E, HKU1, OC43), was reviewed, and the antiviral effect was hypothesis to be inclusive of SARS-CoV-2 [96]. Due to its promising prophylactic effect, a phase 4 clinical trial (NCT04999098) is currently recruiting to evaluate the shedding effect of Enchinaforce against COVID-19. Different forms of Enchinaforce, including Forte tablets, chewable tablets, and tincture, will be introduced to COVID-19-positive participants (Cq < 27), and the differences in Cq value (primary endpoint) comparing pre- and post-Enchinaforce treatment will be evaluated. 

Notably, there have yet to be effective FDA-approved antivirals targeting the SARS-CoV-2 pre-infection stage. Although the previously established entry inhibitor hydroxychloroquine has shown promising in vitro antiviral activity, it has not shown any meaningful in vivo benefits in both animal models and humans [97,98,99,100,101,102,103,104,105,106,107,108,109]. Furthermore, high doses and extended regimens were difficult to achieve due to drug-related severe adverse effects of these drugs [97,99,100,101,102]. Moreover, umifenovir, an antiviral proposed to inhibit SARS-CoV-2 viral entry, has shown some clinical efficacy but is of uncertain clinical importance due to a lack of high-quality evidence and contradicting findings from different clinical studies. Therefore, further investigation into antivirals targeting the pre-infection stage of SARS-CoV-2 is still warranted. The natural compounds discussed in this section can act as pre-infection inhibitors and potentially be utilized as prophylaxis for COVID-19. 

## 5. Post-Infection Inhibitors, Their Proposed Mechanisms, and Their Potency against SARS-CoV-2

Post-infection inhibitors are antivirals that work after a person has been infected with a virus. An effective post-infection drug has the potential to reduce the severity and duration of symptoms and may reduce the risk of complications from the infection. Post-infection antiviral treatment is provided by the physician, specifically for the virus causing the infection, after diagnosis. A significant portion of low-income countries’ populations have been infected with SARS-CoV-2. Thus, effective antivirals for COVID-19 targeting the post-infection stage are essential to prevent the further spread of COVID-19 and mitigate the disease’s impact on the poor.

Shuanghuanglian preparations are TCM extracted from *Lonicera japonica*, *Scutellaria baicalensis Georgi*, and *Forsythia suspense* and are widely used to treat respiratory tract infections [110,111]. Several independent studies revealed that Shuanghuanglian preparations and their active components, baicalin, and baicalein, exhibited potent antiviral activity against SARS-CoV-2 by inhibiting the activity of its 3CL^pro^ and PL^pro^ and its RdRp [53,54]. Due to the high amount of baicalin being present within Shuanghuanglian preparations and almost similar IC_50_ of Shuanghuanglian preparations (1.83 to 6.14 μM) with pure baicalin (6.41 μM), it was believed that baicalin is the main component that led to the proteolytic inhibition of SARS-CoV-2 PL^pro^. The protein–interactions were further evaluated in isothermal titration calorimetry (ITC) measurements as well as electrospray ionization mass spectrometry (ESI-MS), and it was observed that baicalin and baicalein both had a high binding affinity with SARS-CoV-2 PL^pro^ with K_d_ of 11.50 and 4.03 μM in ITC and 12.73 and 1.40 µM in ESI-MS, respectively. The crystal structure of SARS-CoV-2 PL^pro^ complexed with baicalein was further investigated using X-ray crystallography. Results showed that baicalein acted as a shield, interacting with S1/S2 subsites and the oxyanion loop, thus preventing the substrate from binding with the catalytic site of SARS-CoV-2 PL^pro^ [53]. Additionally, baicalein and baicalin were reported to be potent inhibitors of SARS-CoV-2 RdRp. When the SARS-CoV-2 polymerase complex was expressed and purified, the ability of the polymerase to extend the RNA template (14-Mer RNA) in the presence of baicalein and baicalin was investigated. As expected, extended RNA products were reduced significantly in the presence of non-toxic doses of baicalein and baicalin. Molecular docking analysis also confirmed that baicalein and baicalin strongly bind to the RNA polymerase of SARS-CoV-2 with a binding energy of −8.7 and−7.8 kcal/mol, respectively. Therefore, unlike any nucleoside analogs, baicalein and baicalin do not pose a potential risk of host mutational activity, as they directly bind and inhibit SARS-CoV-2 RdRp transcriptional activity. Notably, a pharmacokinetic study in rhesus monkeys revealed that, despite the low oral bioavailability of baicalein, this compound was rapidly converted to the 7-O-βglucopyranuronoside metabolite baicalin. Both compounds demonstrated relatively long half-lives as well as Cmax values (2.3 μM and 30.7 μM, respectively) that were higher than the IC_50_ values (1.2 μM and 6.2 μM, respectively) reported in Calu-3 human lung cells [54], suggesting that less frequent dosage may be needed to maintain therapeutic plasma concentration [112].

A natural extract of *A. paniculate* and its main bioactive compound, andrographolide, were reported to reduce the viral progeny production in plaque assays (IC_50_ = 0.036 μg/mL and 0.034 μM, respectively) and in high-content imaging using an anti-SARS-CoV-2 antibody. The IC_50_ value of *A. paniculate* extract and andrographolide using foci reduction is lower than remdesivir (IC_50_ = 0.086 μg/mL), indicating that they are promising treatment for use against COVID-19 [68]. Furthermore, an in silico molecular docking study suggested that andrographolide might target the main protease of SARS-CoV-2, M^pro^, and absorption, distribution, metabolism, and excretion (ADME) profile prediction showed good pharmacodynamic properties [66]. To further evaluate the interaction of andrographolide and M^pro^, researchers expressed and purified 2019-nCoV M^pro^ and SARS-CoV M^pro^ and reported that andrographolide could form covalent linkages and inhibited proteolytic activities of M^pro^ with IC_50_ of 15.05 and 5.0 μM, respectively. In addition, andrographolide could dock into the catalytic pockets of M^pro^ with a binding affinity from −7.85 to 9.72 kcal/mol [67]. A phase 3 clinical trial (NCT05019326) is currently recruiting 3060 participants to evaluate the efficacy of Andrographis paniculate in asymptomatic COVID-19 with a primary end point defined as the requirement of hospitalization. 

Naringenin is a flavone found in many citrus plants and has various biological functions such as antiviral, antibacterial, and anticancer [113]. Recently, naringenin was identified as a potent antiviral agent against SARS-CoV-2 M^Pro^ (PDB entry: 6w63) from HTVS of 8793 natural compounds. Naringenin and other lead compounds selected from HTVS were reported to inhibit SARS-CoV-2 M^Pro^ in vitro using enzyme inhibition assays (>75% inhibition) at 100 µM. Based on the cell-based antiviral assays, only naringenin showed moderate anti-SARS-CoV-2 activity with IC_50_ of 28.347 μg/mL and SI of 6.3. The structure–activity relationship of naringenin, and its closely related derivative, eriodyctiol, revealed that naringenin (7-OH position) forms H-bonds with Gln189 and Thr190 residues, and its interaction with His41 residues might explain its potent antiviral activity in targeting SARS-CoV-2 M^pro^ in vitro. The molecular properties of naringenin also agreed with Lipinski’s rule. They showed a good score for a topological polar surface area (TPSA) value of <140 Å, indicating that naringenin might have good oral bioavailability, intestinal permeability, and absorption [80]. Parallel with these findings, an in silico study, using molecular docking simulations, molecular mechanics Poisson–Boltzmann surface area (MM-PBSA), density functional theory (DFT), and binding energy analysis supported the interaction of naringenin at the active site of SARS-CoV-2 M^pro^ [114]. 

Resversatrol and pterostilbene were reported to inhibit SARS-CoV-2 progeny production in Vero cells with an IC_50_ of 66 and 19 μM, respectively. These two compounds were also reported to have long-lasting antiviral effects against SARS-CoV-2. They maintained their antiviral activation in cell cultures up to 40 hpi, equivalent to approximately five rounds of viral replications. Time-of-drug-addition assays demonstrated that these compounds solely targeted the viral replication stage but not the entry stage, nor did they have virucidal activity. Additionally, the post-infection antiviral effect of RES was also observed in primary human bronchial epithelial cells cultured under air–liquid interphase (ALI) conditions [81]. Another study also showed evidence that RES could inhibit SARS-CoV-2 (IC_50_ = 10.66 μM) and HCoV-229E (IC_50_ = 4.6 μM) at the post-infection stage with reduced cytotoxicity (CC_50_ = 210 μM) [115]. A cotreatment with silent information regulator T1 (SIRT1) signaling antagonist, sirtinol, and RES attenuated the inhibitory effect of RES. It promoted the replication of the virus, suggesting that RES might inhibit SARS-CoV-2 infection via the SIRT1 pathway [116]. A randomized, double-blind placebo-controlled trial (NCT04400890) reported that resveratrol could lower the incidence of hospitalization, emergency visit, and pneumonia in mild COVID-19 outpatients [117]. However, this clinical trial was limited by a small sample size and low incidence of the primary endpoint (COVID-19 hospitalization). A larger trial is currently undergoing to determine the benefit of resversatrol for COVID-19 and the long-lasting side effects caused by SARS-CoV-2 (post-COVID-19). In a retrospective study NCT04666753, resveratrol was formulated with other natural compounds, including selenium yeast, cholecalciferol, ascorbic acid, ferulic acid, spirulina, N-acetylcysteine and more, to test against SARS-CoV-2 by measuring the clinical symptoms’ duration (primary endpoint). Since resveratrol was reported to prevent liver fibrosis by inhibiting the Akt/NF-κB pathways [118], and COVID-19 causes prolonged fibrotic damage to the lung, a randomized clinical trial (NCT04799743) is currently recruiting to evaluate the anti-fibrotic therapeutic effects of resveratrol on discharged COVID-19 patients. 

## 6. Multi-Stage Inhibitors, Their Proposed Mechanisms, and Their Potency against SARS-CoV-2

Antiviral compounds that can inhibit early and late stages of virus infection are preferable as they have the potential to target multiple stages of the virus life cycle, thus preventing viral adaptation and the emergence of drug-resistant viruses. SARS-CoV-2 is highly mutagenic due to the inherent error-prone nature of RNA replication, and many novel variants of SARS-CoV-2 have emerged. These new variants might confer resistance to antivirals that target a single stage of the virus life cycle. Therefore, the search for SARS-CoV-2 inhibitors that target both the pre- and post-infection stages is crucial in reducing the likelihood of drug resistance. 

Chingguan Yihau (NRICM101) is a TCM formulated in Taiwan and that was used in clinical settings for COVID-19 patients [59]. Patients were administered 100 mL three times daily 30 min after a meal. Clinical data showed that patients with the underlying disease who showed no improvement after 21 days of hospitalizations had benefited from NRICM101 treatment without adverse effects [59]. A large-scale observational study involving 51,000 participants was carried out to evaluate the outcome of NRICM101 on SARS-CoV-2 infection. Participants were subjected to 2–4 times intervention of NRISM101 daily, and the clinical trial’s primary endpoint (NCT04664049) was achieving a negative COVID-19 test result and being free from COVID-19 symptoms within 2 months. The trial has been completed, but the outcomes are still pending. In vitro findings from surface plasmon resonance (SPR) analysis indicated that NRICM101 could bind to the RBD protein dose-dependently. At the same time, ELISA showed that NRICM101 inhibited S protein from binding to the ACE2 receptor with an IC_50_ value of 0.41 mg/mL. Additionally, NRICM101 could also inhibit the 3CL^pro^ enzymatic activity with an IC_50_ value of 0.22 mg/mL. In addition, NRICM101 treatment also reduced the growth of SARS-CoV-2 in terms of viral protein expression (IC_50_ = 0.28 mg/mL) and plaque formation. Besides its promising antiviral activity, NRICM101 could also reduce the production of pro-inflammatory cytokines, IL-6, and TNF-α in lipopolysaccharide (LPS)-stimulated alveolar macrophages with an IC_50_ value of 0.42 and 1.18 mg/mL, respectively [59]. When antiviral studies were performed on the individual herb of NRICM101, researchers reported that the *Scutellaria baicalensis* component was responsible for its effective anti-3CL^pro^ activity (100%). Other components, such as *Schizonepeta tenuifolia*, *Morus alba*, *Magnolia officinalis,* and *Mata* and *Mentha haplocalyx* in NRICM101, effectively blocked the binding of the S protein to ACE2 (>70%). In addition, *Scutellaria baicalensis* and *Houttuynia cordata* with 20-fold dilution could inhibit IL-6 and TNF-α production, and *Scutellaria baicalensis* was able to inhibit cytokine production at 40-fold dilution [59]. Another component of NRICM101, *Mentha haplocalyx,* the natural source of Chinese peppermint, has gained some attention as an antiviral agent against SARS-CoV-2. It was used as traditional medicine for minor ailments [119]. The essential oil extracted from peppermint leaves could improve muscle pain and itching or be used as a fragrance [120]. Using a cell-based HTS assay to screen 190 traditional herbal medicines, researchers identify *Mentha haplocalyx* extract as an effective anti-SARS-CoV-2 compound in vitro and in vivo. Along with *Mentha haplocalyx*, *Ganoderma lucidum* extracts could also reduce the CPE of SARS-CoV-2 in Vero cells at 960-fold dilution. In the hamster model, oral administration of *M. haplocalyx extracts* at 200 mg/kg/day for three consecutive days after viral infection showed promising antiviral activity and significantly reduced lung viral titres [60]. 

Isorhamnetin is a flavonoid extracted from sea buckthorn or *Hippophae rhamnoides* L. The extract from the berries of this plant was reported to have anticancer, antiviral, anti-diabetic, and immune regulatory activities [121,122,123,124]. In one of the studies, the active compound of sea buckthorn, isorhamnetin, was tested against the SARS-CoV-2 S pseudotyped virus in vitro. It was observed that isorhamnetin could block SARS-CoV-2 S pseudotyped virus entry into the cells expressing ACE by 47.7% at a non-toxic concentration of 50 μM. Quercetin, another component of sea buckthorn, was similarly evaluated, but there was no significant reduction in viral entry. Using SPR analysis, isorhamnetin showed an affinity towards the immobilized ACE recombinant protein, and molecular docking analysis further proved that isorhamnetin could bind to ACE2 at K353, E37, and H34 residues [76]. In addition, high throughput virtual screening (HTVS) of naturally occurring phytochemicals also showed that, along with other flavonoids, isorhamnetin could interact with the S2 domain of the SARS-CoV-2 S protein with the binding energy of −8.3 Kcal/mol [78]. On the other hand, various derivatives of isorhamnetin from *Salvadora persica* were reported to have a strong binding affinity towards SARS-CoV-2 M^pro^ with a binding orientation similar to the positive control N3, which binds to the Cys–His catalytic dyad located between domains I and II of SARS-CoV-2 M^pro^. Structure–activity relationship analysis revealed the presence of disaccharide rutinose (α-L-rhamnopyranosyl-(1-6)-β-D-glucopyranose) at position carbon no. 3 flavonoids helped to enhance the binding stability in the N3 binding site of SARS-CoV-2 M^pro^ [77]. Since these studies showed that isorhamnetin could interact with host protein (ACE2 receptor), viral structural protein (S protein), and a viral non-structural protein (SARS-CoV-2 M^pro^), isorhamnetin should be considered for further study to develop it into a potent antiviral drug that could target multiple stages of the viral life-cycle, thus preventing the replication of SARS-CoV-2. 

Ionophore antibiotics are a family of natural compounds produced by microorganisms, which are well known for their antibacterial activity against gram-positive bacteria [125,126]. In addition, the antiviral activity of ionophore antibiotics against HIV, influenza virus, ZIKV, MERS-CoV, and SARS-CoV was also reported [127,128,129]. Recently, researchers screened 11 different naturally occurring polyether ionophores for their potential to prevent the CPE caused by SARS-CoV-2 in Vero E6 cells that overexpressed TMPRSS2. Ionophore antibiotics such as narasin, salinomycin, and nanchangmycin exhibited >100-fold selectivity. Surprisingly, two compounds, maduramycin and X-206, showed a selective index of 313 and 586, respectively, higher than the remdesivir control (SI ≥ 67). Compound X-206 displayed significant antiviral activity against SARS-CoV-2 by reducing CPE (IC_50_ = 14 nM), viral RNA copy number, viral S proteins, and viral plaque formation by SARS-CoV-2. Time-of-addition assays revealed that X-206 achieved approximately two-fold log reduction in viral progeny at all time points from 4 h pre-infection to 8 h post-infection. Based on morphological profiling, which provided bioactivity fingerprints, X-206 had a different mechanism of action than the control lysosomotropic hydroxychloroquine (HCQ). However, the molecular mechanism of this compound has not been elucidated, and further experiments should be conducted to discover the mode of action of X-206 [86]. The ionophore antibiotics, salinomycin and niclosamide, were observed to inhibit syncytia formation, which indicates that this is caused by the fusion of cells induced by viral infection. It was observed that niclosamide and salinomycin exhibited cell protection effects against SARS-CoV-2 with IC_50_ of 0.34 and 0.22 μM, respectively. Additionally, salinomycin and niclosamide also inhibited viral replications in respiratory Calu-3 cells. Furthermore, a low concentration of 1 μM of niclosamide blocked the transmembrane member 16 (TMEM16) chloride channel. It significantly attenuated the effect of spontaneous calcium transients in the presence of the S protein, suggesting that ionophore antibiotics might block intracellular calcium release to prevent syncytia formation [87]. Unfortunately, a small scale (73 participants) phase 2 randomized clinical trial (NCT04399356) reported that niclosamide intervention did not shorten the symptom duration of mild to moderate COVID-19 compared to the placebo group [130]. A larger (1200 participants) phase 4 randomized clinical trial (NCT05087381) is currently ongoing in order to further evaluate the benefit of niclosamide for COVID-19 early treatment.

Oleandrin is one of the bioactive compounds found in *Nerium oleander* extract and was identified as a unique lipid-soluble cardiac glycoside that acted on the Na/K ATPases pump to enhance heart contraction in heart failure patients. It was also reported to be used in treating dermatological diseases and cancers [131,132,133]. Recently, oleandrin was reported to be a potent inhibitor of SARS-CoV-2. When oleandrin was present before (prophylactic) and during the whole viral life cycle, it could completely reduce the plaque formation of SARS-CoV-2 by 4 log_10_, with IC_50_ values of 11.98 and 7.07 ng/mL for 24 and 48 hpi, respectively. Additionally, when oleandrin was added 24 h post-infection, where extensive viral replication had already occurred, oleandrin could still maintain its therapeutic effect. In the hamster model, oleander extract did not cause any toxicity effects regarding body weight, organ lesions, alkaline phosphatase (ALP), and alanine aminotransferase (ALT) levels. In vivo, the prophylactic efficacy of oleander extract against SARS-CoV-2 showed a significant reduction in viral loads in nasal turbinates after 3 days post-infection. From day 3 onwards, the viral titre was below the detection limit and, after four days of treatment, the virus was completely cleared [64]. Besides SARS-CoV-2, oleandrin was also reported to be a potent inhibitor of HCoV-OC43 when CPE (IC_50_ = 26 mM) and viral titre (2–3 log10 reduction) were measured as the end point [134].

## 7. Natural Products with Immunomodulatory Effects

An effective immune response against COVID-19 requires both an innate immune system and an adaptive immune system. Studies have reported on the role of innate immunity in determining the severity and prognosis of COVID-19. As the first line of defense against SARS-CoV-2, innate immune cells secrete interferons and other cytokines upon virus entry, thereby preventing virus entry, replication, and assembly, eliminating infected cells, and accelerating adaptive immunity [135]. SARS-CoV-2-specific CD4^+^ T cells, CD8^+^ T cells, and antibodies produced by B cells are the key components of adaptive immunity against COVID-19. Several research groups have reviewed the roles of adaptive immunity and their interplay with innate immunity in COVID-19 [136,137]. Upon 5–6 days after contact, most SARS-CoV-2 infections in patients present with mild symptoms, such as fever, dry cough, tiredness, sore throat, loss of taste and smell, and skin rashes [138]. However, in some cases, the disease may progress to life-threatening pneumonia with a high risk of developing progressive respiratory failure that may lead to death, particularly in older patients or those with underlying comorbidities [139]. Attenuating immunopathogenesis of COVID-19, such as cytokine storm and excessive immune cell infiltration and suppressing viral replications, are considered important hallmarks in halting the progression to severe disease and reducing mortality. Corticosteroid usage in severe COVID-19 patients was shown to be beneficial in reducing mortality and attenuating excessive inflammation [140,141,142]. Nevertheless, they carry several disadvantages, including immunosuppression [143], increased risk of secondary bacterial infections [141], increased risk of developing systemic complications [144], and prolonged viremia [141,145]. Additionally, early corticosteroid treatment was associated with an increased risk of death in COVID-19 patients, particularly those below 60 years old who were not presenting inflammation upon admission [140]. 

EGYVIR extract containing *Curcuma longa* (Turmeric) root and *Piper nigrum* seed extracts have demonstrated immunomodulatory properties and potent antiviral activity against SARS-CoV-2. EGYVIR reduced the viral plaque formation by up to 92% with an IC_50_ of 0.57 μg/mL. When treatment was performed 24 h post-infection, EGYVIR extract remained effective in inhibiting the virus infectivity by 78%, which was higher than the control, hydroxychloroquine (66%). Additionally, the extract was found to downregulate the expression of inflammatory markers such as Ikβα, TNF-α, and IL-6 in SARS-CoV-2 infected Huh7 cells, suggesting that it might alleviate the cytokine storm induced by SARS-CoV-2 infection [63]. Similarly, TCMs such as Lianhuaqingwen (LHQW) [56], Liu Shen (LS)[55], and Chingguan Yihau [59] have been shown to suppress SARS-CoV-2 replication and excessive cytokine production in vitro. LHQW has been shown to inhibit SARS-CoV-2 replication with an IC_50_ of 411.2 μg/mL. Under TEM, LHQW-treated cells (48 h) showed reduced virions in the cytoplasm, endoplasmic reticulum, cell membrane, and intracellular vesicles. LHQW treatment was also observed to alter SARS-CoV-2 morphology. Moreover, LHQW inhibited SARS-CoV-2-induced cytokine production in vitro, as shown by significantly reduced mRNA expression levels of tumor necrosis factor-alpha (TNF-α), interleukin 6 (IL-6), C-C motif chemokine ligand 2 (CCL-2)/monocyte chemoattractant protein 1 (MCP-1), and C-X-C motif chemokine ligand 10 (CXCL-10)/interferon gamma-induced protein 10 (IP-10) [56]. In a prospective, multicenter, open-label, randomized controlled trial of 284 hospitalized COVID-19 patients, treatment with LHQW improved the recovery rate of chest computed tomography (CT) manifestations, shortened the median time to the symptom (fever, cough, and fatigue) recovery, and enhanced the rate of symptom recovery [146]. Two meta-analyses revealed that the integration of LHQW with Western medicines compared with Western medicine alone also led to significantly improved CT imaging, a reduced rate of severe disease progression, and an improved recovery and disappearance rate of clinical symptoms such as fever, cough, and fatigue [147,148]. More recently, the China FDA approved the use of LHQW as a therapeutic agent against COVID-19 due to ample clinical findings substantiating its immunoregulatory effect in COVID-19 patients. Another TCM, Liu Shen (LS), besides its direct antiviral effect (IC_50_ = 0.06024 μg/mL), could downregulate the nuclear factor kappa-light-chain-enhancer of activated B cell/mitogen-activated protein kinase (NF-κB/MAPK) signaling pathway, which was reflected in the decrease in expression of p65, of phospho (p)-nuclear factor of kappa light polypeptide gene enhancer in B-cell inhibitor alpha (IκBα), and of p-p38 MAPK. The downregulation of the NF-κB/MAPK signaling pathway was postulated to decrease pro-inflammatory cytokines such as IL-6, TNF-α, IL-1β, CXCL10/IP-10, CCL2/MCP-1, and IL-8, suggesting that it may prevent cytokine storm in COVID-19 [55]. Another TCM, Chingguan Yihau, formulated in Taiwan, was used in clinical settings for COVID-19 patients. Clinical data showed that patients with underlying diseases who showed no improvement after 21 days of hospitalization benefited from treatment without adverse effects. The mechanism of action of Chingguan Yihau was reported to be through its ability to reduce the production of proinflammatory cytokines, IL-6, and TNF-α, with an IC_50_ value of 0.42 and 1.18 mg/mL, respectively, in lipopolysaccharide (LPS)-stimulated alveolar macrophages [59]. Network pharmacology analyses revealed that LHQW [149,150], Jinhua Qinggan [151], and Xuebijing injection (XBJ) [152] could potentially treat COVID-19 by modulating the various genes and signaling pathways involved in anti-inflammatory, antiviral, and immune responses. Additionally, molecular docking analyses of the bioactive constituents of these TCMs were shown to bind with various viral and host proteins, including Mpro, 3CL^pro^, S protein, and ACE2 [149,150,151,152,153], which, together with the previous findings, suggest a multitude of mechanisms of TCMs in ameliorating COVID-19 infection and immunopathogenesis.

Some COVID-19 patients were also reported to have gastrointestinal microbiota dysregulation. Over time, probiotics have been reported to improve general health by preventing gastrointestinal dysbiosis and helping with gastrointestinal, oral, and vaginal infections [154,155]. Moreover, the immunomodulatory effects of probiotics have gained attention concerning their anti-allergy and antiviral activities [156]. Antigens from the probiotics, such as lipoteichoic acids (LTA), lipopolysaccharides (LPS), and bacterial nucleic acid, could bind to specific host cell receptors such as Toll-like receptors (TLR) and give rise to immunomodulatory effects [157,158]. Moreover, probiotics may be more accessible and affordable than synthetic drugs for COVID-19 in low-income countries for several reasons. Firstly, probiotics are generally less expensive than synthetic drugs, making them more affordable for people living in low-income countries. Secondly, probiotics are widely available in many regions, including low-income countries. This means that people living in areas with limited access to synthetic drugs may be able to use probiotics as an alternative treatment option. Lastly, probiotics are generally regarded as safe. They have fewer side effects than synthetic drugs, which can be particularly important in areas where access to medical care may be limited [159]. Various probiotics, especially lactic acid bacteria (LAB), have been reported to modulate the immune system and prevent viral infection. For example, live and heat-killed *Enterococcus faecalis* KH2 could protect animals against influenza virus and EV-71 by modulating monocyte chemoattractant protein-1(MCP-1), a key chemokine that regulates migration and infiltration of immune cells to the site of infection. A pretreatment with *E. faecalis* reduced the MCP-1 production, thus preventing the migration, infiltration, and differentiation of monocyte, preventing inflammatory disorder, reducing lung viral titer, and enhancing the survival rate of influenza-infected mice. Furthermore, a pretreatment with *E. faecalis* reduced the MCP-1 and protected mice against EV71-induced neurotoxicity and growth retardation in EV-71-infected mice [160]. LAB is also well known for altering immune cells’ functional role, especially dendric cells. For example, *Bifidobacteria* strains in clinically active probiotic combination (VSL#3) could stimulate intestinal and blood DC to upregulate IL-10 production and downregulate the expression level of costimulatory molecules CD80 and CD40, thus preventing the generation of pro-inflammatory Th1 cells and decreasing IFNγ production by T cells [161]. In addition, LC-plasma could also stimulate pDC to produce type 1 IFN via TLR9 and subsequently induce the upregulation of ISGs in HepG2 cells, thus inhibiting the replication of DENV [162,163]. Based on these studies, probiotics or LAB might have a good advantage in reducing COVID-19 severity and in speeding up viral clearance. In an open-label propensity-score-matched retrospective cohort study, the probiotics Bifidobacterium, Lactobacillus, and Enterococcus could reduce time to clinical improvement, days of hospitalization, and time to viral RNA clearance of COVID-19 [164]. *Lactiplantibacillus plantarum* is a probiotic that was reported to modulate cytokine production by increasing TNF-α, IL-1β, IL-18, and IL-8, and reducing IL-6, a critical factor in immune dysregulation in COVID-19, thus enhancing the activity of natural killer cells [165]. In a quadruple-blinded, randomized, placebo-controlled trial, *Lactiplantibacillus plantarum* (KABP022, KABP023, KAPB033) and *Pediococcus acidilactici* (KABP021) could shorten the disease recovery period and symptom duration [166]. Recently, *Bacillus subtilis natto*, a probiotic used to ferment soy food, was reported to stimulate human M1-phenotype macrophages by upregulating immune activation markers, CD80, and enhanced the gene expression of interferon stimulative genes (IFNβ1, IFNλ1, ISG20, and RNase L), thus was suggested to be used as an antiviral agent against SARS-CoV-2 [167]. A double-blinded, randomized, placebo-controlled trial was performed with three strains of bacillus, *Bacillus coagulans* LBSC (DSM 17654), *Bacillus subtilis* PLSSC (ATCC SD 7280), and *Bacillus clausii* 088 A E (MCC 0538), concerning COVID-19 symptoms. The results from this trial showed that the mental and physical fatigue score of the treatment group were significantly improved (91%) [168]. Based on the evidence mentioned above and the absence of conclusive treatments for COVID-19, probiotics supplementation is highly anticipated by the public, especially the socioeconomically challenged population, because probiotics are generally recognized as safe (GRAS), low in cost, and are easily found in daily diets. However, further preclinical and clinical studies should be explored to discover the role of probiotics in COVID-19. Many clinical trials investigating the benefits of probiotics for COVID-19 are ongoing.

## 8. Application Strategies

### 8.1. Combinational Therapy of Natural Compounds

Researchers have shown interest in exploring the potential of natural products as anti-SARS-CoV-2 agents. However, research on their combined effectiveness with other antiviral agents is lacking. Combination therapy can help target multiple viral life cycles to promote effective antiviral effects, which far outweigh simple monotherapy. Combination therapy, which involves two or more therapeutic agents, has been a cornerstone of antiviral therapy. For example, combination treatment involving at least three different antiretroviral drugs (two nucleoside reverse transcriptase inhibitors, and one either an integrase strand transfer inhibitor (INSTI)—a non-nucleoside reverse transcriptase inhibitor—or a protease inhibitor) is recommended for the standard treatment of HIV [169]. Such a combination therapy could inhibit HIV replication, allowing the immune system to overcome the infection and prevent AIDS. Additionally, the repurposing drugs, sofosbuvir and daclatasvir, when combined, could synergistically inhibit SARS-CoV-2 replication by targeting different processes during viral RNA synthesis, and this combination has shown promising efficacy in several clinical studies [170,171,172]. Thus, it may be an interesting avenue worth exploring, considering the promising preliminary evidence suggesting the potential benefit of natural products for COVID-19. It is plausible that the combinatorial use of various natural products or natural products with other anti-SARS-CoV-2 therapies (monoclonal antibodies, small chemical molecules, antiviral peptides, and repurposing drugs) could lead to better clinical outcomes in severe COVID-19 and achieving better survivability, which warrants further investigations. For example, a combination of NhPV/Suramin has been shown to synergistically inhibit SARS-CoV-2 viral entry when used with a SARS-CoV-2 neutralizing antibody (SAD-S35) [75]. Intriguingly, some of the aforementioned natural products have reported various drug–drug interactions, which, if used in combination, might lead to summation (additivity), enhancement (synergism), or initiation (potentiation) of antiviral effect against COVID-19. For example, pomegranate extract has been studied for its potential use with other antiviral compounds. A study has shown that a pomegranate rind extract (PRE) and punicalagin, when co-administered with zinc (II) ion, were more effective at inhibiting the replication of the acyclovir-resistant HSV than either compound alone [173]. Additionally, PUG has also been shown to have synergistic antiviral effects with oseltamivir, an antiviral drug to which the human influenza A virus has conferred resistance [174]. Furthermore, the impact of PUG combined with zinc sulfate monohydrate on SARS-CoV-2 3CLpro was reported, and the coefficient of drug association (CDI) was 0.272, indicating the synergistic effect of combining these two compounds [175]. Another compound, quercetin, present in APRG64, LHQW, XBJ, and *Vitis vinifera* leaf extract, was identified as an anti-SARS-CoV-2 compound and was reported to have a synergistic antiviral effect when combined with vitamin C. The synergistic effect of quercetin and vitamin C was believed to be attributed to the potent antiviral effect of quercetin, the immunomodulatory effects of vitamin C, and the capacity of vitamin C to recycle quercetin [176]. Natural products such as natural extracts or TCMs are considered combination therapy. They comprise diverse bioactive constituents, allowing synergistic interactions between each ingredient and exerting multiple mechanisms, such as immunomodulatory, anti-inflammatory, and antiviral responses. For example, when the active compound of Aetemisia annua L. extract, artemisinin, was tested as a pure drug, it was less effective (IC_50_ = 70 μM) than the Aetemisia annua L. cultivar extracts (IC50 = 0.01 to 0.14 μg/mL), suggesting that artemisinin might not be the sole active ingredient contributing to the anti-SARS-CoV-2 activity of this extract [62]. For example, a mixture of polysaccharide 375 isolated from the seaweed Ecklonia kurome Okam, was more effective in inhibiting the enzymatic activity of SARS-CoV-2 M^pro^ when compared to purified homogenous polysaccharides 37501 and 37503. This finding suggested that the cocktail-like polysaccharide synergistically inhibited SARS-CoV-2, supporting that it might be a potential antiviral drug for COVID-19 [83]. Since low-income countries face several challenges in combating COVID-19, combining antiviral therapy constituted by natural products can offer several advantages. Firstly, a combination therapy using natural compounds might improve treatment outcomes by reducing the risk of treatment failure. Ensuring the effectiveness and sustainability of antiviral treatments would potentially reduce the need for future hospitalizations and other medical interventions. Secondly, combining multiple drugs in a single regimen might reduce the risk of drug resistance, thus reducing the cost to develop new antiviral drugs. This could make it more feasible for low-income countries to provide treatment to a larger portion of the population. When used in conjunction with other public health measures such as vaccination and infection control, over time, combination therapy can help to reduce the burden of viral infections in low-income countries. However, further in-depth studies to evaluate the efficacy and safety of combined natural products against COVID-19 should be prioritized in the current global health pandemic.

### 8.2. Potential Delivery Methods of Natural Compounds

To reduce the viral load in COVID-19 patients, it is crucial to inhibit viral replication and virus production by host cells. Although various antiviral compounds may be active in the free form, their ability to concentrate at the sites of infection may be limited, requiring the administration of a high dose with accompanying adverse effects. Thus, an effective delivery system using nanocarriers such as dendrimers, liposomes, and polymers to transport antiviral drugs against SARS-CoV-2 may be an added advantage [177]. Antiviral natural compounds can be entrapped or coupled with these nanocarriers to enhance their solubility, stability, and bioavailability [177]. For example, polyamidoamine (PAMAM) dendrimer conjugation could enhance the inhibitory effect of sialyl lactose against a wide range of human and avian influenza viruses [178]. In addition, PAMAM dendrimer is effective in pulmonary delivery of the poorly soluble anti-asthma drug, beclometasone dipropionate. Since the main SARS-CoV-2 tropism occurred in the respiratory system, aerosol-based PAMAM dendrimer might be an excellent delivery system for natural antiviral compounds against COVID-19. Liposomes consisting of mainly amphoteric phospholipids bilayer could effectively enhance the delivery of water-soluble and lipid-soluble antiviral compounds, thus enhancing the bioavailability of the compound [179]. Long-circulating-nanoliposome (LCNs)-containing baicalein (BAI) have been proven to yield greater LCNs-BAI oral bioavailability compared to free BAI by 4.52 times in Kunming mice [180]. Since baicalein was reported to be an effective antiviral against various viruses, including SARS-CoV-2, this study provides a beneficial and promising therapeutic platform for baicalein for COVID-19 when incorporated with liposome technology. In another study, liposome formulation could enhance the antiviral activity of propolis in inhibiting the 3CL protease (IC50 = 1.183 ± 0.06) compared to free-form propolis (IC50 = 2.452 ± 0.11) and significantly reduced the viral replication of SARS-CoV-2 with an inhibitory effect comparable to remdersivir [181]. On the other hand, liposome was also suggested to be an effective delivery system for TCM to improve targetability and reduce side effects [182]. A recent study has shown that tripterin, a TCM, was successfully encapsulated into liposomes and significantly reduced the pulmonary pathological alterations and viral burden of severe COVID-19 in the human ACE2-transgenic (hACE2) mice model [183]. Next, polymeric nanoparticles such as chitosan are also valuable for developing nanoformulations for COVID-19. Studies have suggested that amphiphilic positively charged chitosan might electrostatically bind to the virus and prevent viral entry into the host cells [184]. Besides having potent anti-SARS-CoV-2 activity, N-palmitoyl-N-monomethyl-N,N-dimethyl-N,N,N-trimethyl-6-O-glycolchitosan (GCPQ) has been reported to reside (13.1%) in mouse nares up to 24 h after nasal dosing, suggesting that it could use as a prophylactic nasal spray to prevent COVID-19. Since chitosan has good antiviral activity, it may be used to encapsulate antiviral drugs to achieve combinational effects, thus developing a comprehensive antiviral therapy for COVID-19. Various studies have proven that using chitosan as an effective delivery system positively affects the physiological function of natural compounds. For example, punicalagin-loaded alginate/chitosan-gallol hydrogels enhanced tissue adhesiveness and promoted wound repair and hemostasis [185]. Recently, baicalein-encapsulated chitosan has been successfully developed and has achieved high loading (74.2%) and encapsulation (96.1%) capacity. The delivery system was also reported to provide a more controlled and constant effect on tissue targets without adverse effects [186]. Although the antiviral effect of baicalein-loaded chitosan has not been tested, it has been reported that such a delivery system could effectively control immune-allergic-inflammation in asthmatic mice [186]. On the other hand, EGCG-loaded silver nanoparticles (EGCG-AgNPs), combined with zinc sulfate, could exert a triple effect on multiple steps of the infectious cycle of H5N1 [187]. This strongly demonstrated that co-administering zinc sulfate with a natural compound and using silver nanoparticles as a delivery system might be a potential platform for developing SARS-CoV-2 inhibitors, which can prevent microbial resistance by restricting adaptation. Furthermore, advanced modifications of these nanocarrier-coated antiviral drugs are feasible to reduce the potential adverse effects or elevate the draggability of the antiviral compounds. For example, the surface modification of cationic dendrimers to becoming more negatively charged is directly linked to reduced cytotoxicity and may be used to transport antiviral compounds against SARS-CoV-2 [188]. Moreover, coupling polyethylene glycosylated (PEG) to nanocarrier surfaces could prevent the binding of plasma proteins to the nanocarriers, avoid degradation by the immune system [189], enhance the half-life, and extend the circulation time of these antiviral compounds [190]. For instance, PEGylated PAMAM dendrimer has been reported to improve the drug loading capacity of the anti-HIV drug, Lamivudine and reduce the high surface toxicity of PAMAM while helping to control the release of antiviral drugs in a prolonged manner [191]. Furthermore, antiviral-compound-containing nanocarriers may be coated with SARS-CoV-2 antibody to direct them to the infection site, thus effectively targeting infected cells. This approach has been constructive in anticancer treatment when antibody-modified PAR28-PEG-liposome effectively delivered the drug into the targeting cells [192]. Overall, a powerful antiviral delivery system may promote drug disposition at the sites of infection, enhance intracellular drug delivery, prevent drug inactivation, and prevent toxicity caused by overdosing. As a result, incorporating nanotechnology with natural compounds should be the prospective research topic for COVID-19 drug discovery.

### 8.3. Potential Route of Administration

Promising natural compounds have been proposed to inhibit SARS-CoV-2, but the draggability of these antiviral compounds remains the most challenging task. An effective antiviral should achieve a concentration level within the therapeutic range at the infection site to produce the desired clinical outcome. As a result, the administration route is one way to ensure adequate drug concentration at the targeted organ while avoiding toxicity (overdose) and ineffective treatment (underdose). Although intravenous administration of antiviral drugs is frequently applied, it has various drawbacks. For example, it requires trained personnel, specialized equipment, and consumables, and it causes mild discomfort to the patient. Due to the intravenous use of remdesivir in COVID-19 patients, it is limited to hospital settings. Poor levels of health service coverage, low levels of government funding, and a lack of political support are some of the causes of inadequate response to injecting antiviral drugs in low-income countries [193]. Recently, inhalation treatment using a portable inhaler device was suggested for COVID-19 [194] as it is convenient and easily accessible, which could provide early doses of antiviral, thus curtailing the development and transmission of COVID-19. Since the SARS-CoV-2 infection is spread through airborne virions, the ciliated nasopharyngeal epithelium and oral cavity were believed to be the primary site of infection [195]. Studies have reported that the ACE2 receptor is highly expressed in the nasal cavity compared to peripheral lung tissue [195], supporting that the nasal cavity is the primary portal for the virus to internalize into the human body [196,197]. As a result, applying an antiviral nasal or oral spray might reduce the infectious virus load and the risk of transmission. Recent research has demonstrated that nasal sprays could inhibit several respiratory viruses’ replication [198]. For example, AM-301 or Bentrio, a nasal spray containing bentonite (magnesium aluminum silicate), was demonstrated to reduce the viral load in human airway epithelial cells without causing toxicity to the nasal epithelium [199]. Next, povidone-iodine (PVP-I) nasal antiseptic preparation was reported to have strong virucidal activity against extracellular SARS-CoV-2 [200]. The efficiency of PVP-I nasal sprays in reducing SARS-CoV-2 nasopharyngeal titer in the clinical setting was also evaluated (NCT04347954). Moreover, when the cocktail of F61 and H121 neutralizing antibodies were administered via the intranasal route, this significantly improved the survival rate and reduced the lung virus of delta-SARS-CoV-2 (B.1.617.2)-challenged K18-hACE2 mice [201]. Thus, nasal drops or sprays might be an excellent route for administering prophylactic natural compounds. The nasal spray of grapefruit seed extract, which contains bioactive metabolites such as limonin, naringenin, and hesperidin, has been commercialized and was suggested by physicians to be used for COVID-19 in case series [202]. In addition, Product I, an oral spray containing a mixture of natural essential oils (Anise oil, eucalyptus oil, levomenthol, myrrh extract, clove oil, peppermint oil ratanhia root extract, and tormentil root extract), showed potent virucidal activity and reduced the virus load to lower limit of detection (≥4.69 log10 TCID50/mL), compared to other chemical sprays [203]. The repurposed herbal spray, Allergic Rhinitis Nose Drops (ARND), containing mainly Herba Centipedae, mint, and *Scutellaria baicalensis*, has been reported to inhibit pseudovirus infection, modulate the immune system, and prevent binding between the spike protein (Delta) and ACE2 of SARS-CoV-2 [204]. These preliminary data of commercialized natural nasal or oral sprays provide the basis for further clinical investigations and future development of the antiviral hygienic product for COVID-19. Although nasal sprays are an effective tool in protecting from and preventing initial contact with SARS-CoV-2, they might not be effective when SARS-CoV-2 spreads further to the lower respiratory tract. In this respect, an oral inhalation nebulizer is an alternative tool to concentrate the antiviral drugs in the lung and prevent pneumonia, acute respiratory distress syndrome (ARDS), sepsis, and even superinfection [205,206]. An inhalation nebulizer is suitable for antiviral compounds with low oral bioavailability and a shorter half-life [207]. An inhalation of antiviral compounds into the lung means that they are more readily absorbed and distributed systemically compared to oral dosing, as the latter depends on gastrointestinal tract (GIT) absorption and systemic metabolism in GIT and the liver [208,209,210]. For instance, when budesonide drug, with high first-pass metabolism and low GIT absorption properties, was administered through inhalation, it reduced the likelihood of intensive care and the recovery period in COVID-19 patients [208]. On the other hand, a natural essential oil blend made up of lemon oil, oregano oil, tea tree oil, java citronella oil, turmeric oil, peppermint oil, lavender, ginger, frankincense, eucalyptus, wheat germ, basil, cedarwood, holy basil, cinnamon oil, sage oil, and clove oil (registered batch no. 250320211) was reported to be safe and could be used to relieve respiratory symptoms of mild COVID-19 patients, when administered via nebulization [209]. Unfortunately, using nebulizers is concerning as the aerosol particles generated by nebulizers are small (1–5 μm) and can carry contaminants such as viruses and bacteria into the deep lung, thus increasing the risk of COVID-19 transmission [210]. As a result, care should be taken when using an antiviral nebulizer [211]. Figure 3 summarizes the therapeutic applications of antiviral compounds against SARS-CoV-2.

## 9. Conclusions

The SARS-CoV-2 pandemic represents an unprecedented crisis to healthcare facilities and socioeconomics worldwide. The urgent need for effective treatment for COVID-19 has led to the extensive repurposing of drugs as a convenient strategy in identifying antiviral agents as a rapid treatment response to this global crisis. However, despite many drugs showing promising preclinical efficacy against SARS-CoV-2, most have failed to translate into clinical benefits in humans. Although remdesivir has been FDA-approved for treating COVID-19, it does not cure or prevent the spread of COVID-19, given its modest overall efficacy. Furthermore, the use of many antiviral drugs, including remdesivir, is only limited to hospital use, and low-income countries commonly do not have the privilege to afford such treatment as they are facing challenges such as lack of healthcare providers, personal protection equipment, healthcare system, financial leverage, and more. As a result, the search for new effective, widely accessible, affordable, and broad-spectrum antiviral drugs against SARS-CoV-2 is still warranted. Much work has been dedicated to using in silico, in vitro, ex vivo, and in vivo methods to predict, evaluate, test, and develop efficient antiviral treatments for SARS-CoV-2. An in silico study uses computer simulations and modeling software to predict the compounds’ behavior in biological systems. Useful molecular docking software such as AutoDock Vina, UCSF DOCK6, Glide, and Surflex-Dock allow researchers to perform high-throughput screening to determine the binding affinity of potential compounds to SARS-CoV-2 proteins. However, in silico software rely on assumptions and simplifications that may not accurately reflect the complex biological systems. For example, the proteins may be modeled as a rigid structure rather than flexible and dynamic molecules, or the ligand affinity towards the proteins was assumed based on empirical data, known as force fields. Although in silico studies provide valuable information on the potential of the antivirals, it is important to validate the predictions made by in silico models using in vitro and in vivo studies. While in vitro studies have proven helpful in selecting antivirals, it is important to acknowledge their limitation in mimicking the physiological environment. Some in vitro systems, such as organoids or microfluidic chips, can better mimic living organisms’ complex cellular and physiological interactions. Furthermore, since the in vitro findings in this review were collected from different labs, the results should be interpreted with caution, and the potency of other natural antivirals may not be directly compared but only used as a reference because the experimental conditions, settings, and techniques used may differ. Taken together, nature is an excellent reservoir of potent antiviral compounds that can be directly used or modified into new drugs. This review introduced some promising natural products, including TCM, natural extracts (or their active compound), and probiotics, all with reported activity against SARS-CoV-2, and further discussed their efficacy and mechanism. Although most of the natural products listed in this study are potential inhibitors of SARS-CoV-2, the results are primarily derived from in vitro research. Thus, future work should critically evaluate the bioavailability, suitable dosage, mode of delivery, therapeutic window, and combinatorial potential of these natural products in animal models before advocating their use as proof-of-concept antiviral agents in human COVID-19 clinical trials. Additionally, future efforts should focus on improving the draggability and efficacy of potential and existing antiviral drugs for COVID-19 treatment. Such measures might include developing aerosol-based antivirals to be administered via inhalation, allowing rapid accumulation in the lungs where SARS-CoV-2 viruses mainly infect, or utilizing a nano-carrier delivery system to enhance targeted drug delivery and bioavailability. In summary, while natural products for COVID-19 have shown promise, further research is needed to fully understand their efficacy and safety, particularly in different populations and in conjunction with new treatments.

## Figures and Tables

**Figure 1 ijms-24-09589-f001:**
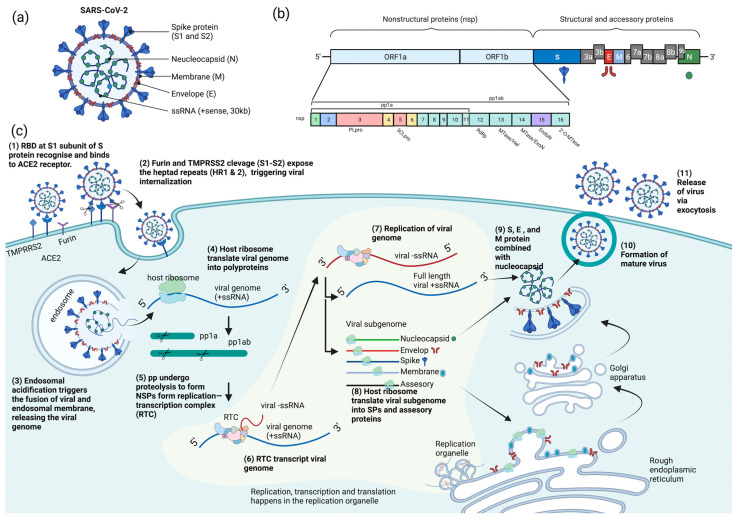
Schematic diagram of SARS-CoV-2 (**a**) structure, (**b**) genome encoding for structural, non-structural, and accessory proteins, and (**c**) replication cycle. ACE2 = angiotensin-converting enzyme 2, pp = polyprotein, RBD = receptor-binding domain, RTC = replication and transcription complex, SPs = structural proteins, ssRNA = single-stranded RNA, TMPRSS2 = transmembrane serine protease 2, S = spike, N = neucleocapsid, M = membrane, E = envelop, NSPs = nonstructural proteins, PL^pro^ = papain-like protease, 3CL^pro^ = chymotrypsin-like protease, RdRp = RNA-dependent RNA polymerase, MTase = methyltransferase, ExoN = exonuclease, and EndoN = endonuclease (figure drawn by biorender).

**Figure 2 ijms-24-09589-f002:**
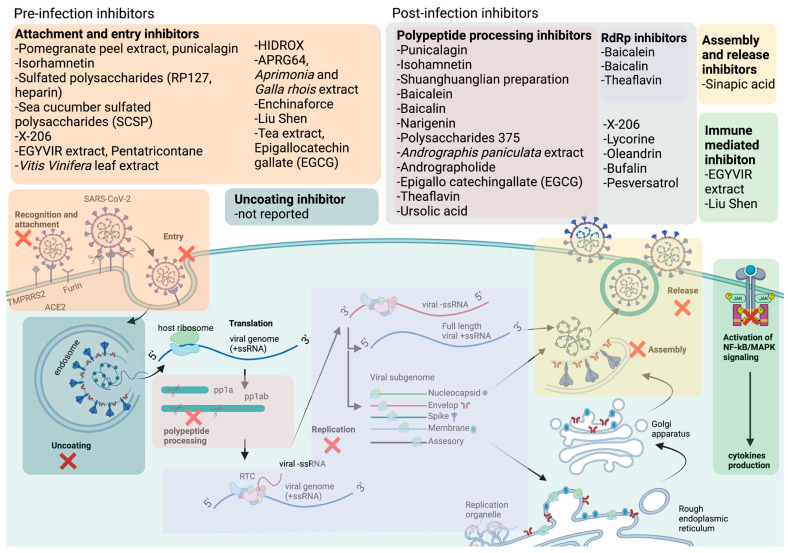
The antiviral activity of natural compounds, including natural extracts and their active ingredients, in the life cycle of SARS-CoV-2. Natural compounds reported against SARS-CoV-2 were categorized according to the mechanism of inhibition at different stages of the SARS-CoV-2 life cycle. Natural compounds affecting uncoating have not been reported. RTC = Replication–transcription complex, +ssRNA = positive single-stranded ribonucleic acid, -ssRNA = negative single-stranded ribonucleic acid TMPRRS2 = Transmembrane serine protease 2, ACE2 = angiotensin-converting enzyme 2, pp = polypeptide, NF-kB = Nuclear factor-kB, MAPK = Mitogen-activated protein kinase, PPE = pomegranate peel extracts, PUG = punicalagin EGCG = epigallocatechin gallate LHQW = Lianhuaqungwen, JHQG = Jinhua Qinggan, XBJ = Xuebijing. (Figure drawn by biorender).

**Figure 3 ijms-24-09589-f003:**
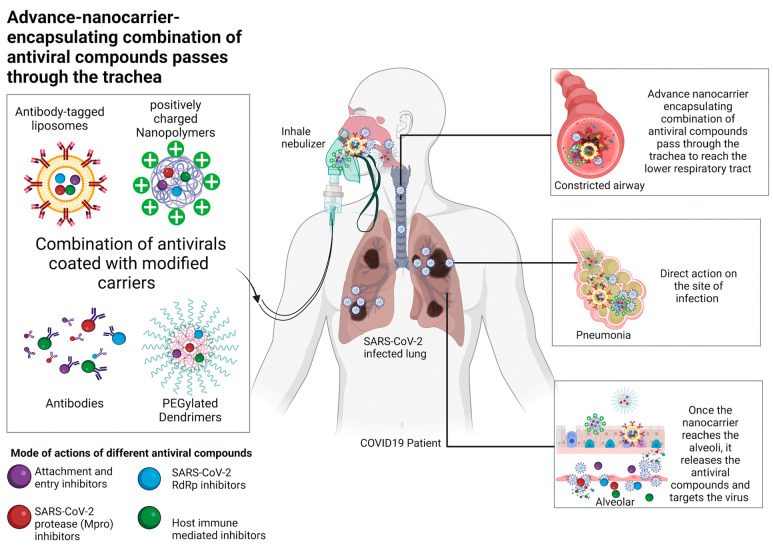
Therapeutic applications of antiviral compounds against SARS-CoV-2. Different carriers such as liposomes, polymers, dendrimers, and antibodies can deliver antiviral compounds via the inhaling nebulizer route, concentrating the antiviral compounds in the lung where SARS-CoV-2 mostly resides. Modifications such as antibody tagging, polyethylene glycol PEGylation, and surface charge alteration can advance the pharmacodynamics of these antiviral compounds. Various antiviral compounds with different modes of action can be simultaneously encapsulated in a carrier to reduce toxicity and prevent the development of antiviral-resistant mutants. (Figure drawn by biorender).

**Table 1 ijms-24-09589-t001:** Examples of natural compounds against SARS-CoV-2. (Compound structures are drawn using ChemDraw Professional 16.0).

Natural Compounds	Compound Structure	Genotype/Strain	IC_50_	CC_50_	Stages of Inhibition	Suggested Mechanism	RCTs	References
		Traditional Chinese Medicine (TCM)						
Shuanghuanglian preparations	NA	2019-nCoV	0.064–0.090 µL/mL ^A,1^ 0.010 mg/mL ^A,2^ 3.65–4.44 µL/mL ^A,3^ 0.14 mg/mL ^A,4^ 0.93–1.20 μL/mL ^B^	>12.5 μL/mL	Polypeptide processing and replication	Inhibited catalytic activity 3CL^pro^ Inhibited polymerization of RdRp.	NA	[53,54]
Baicalein	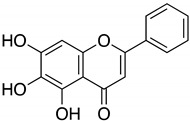		0.94 μM ^A^ 2.94 μM ^B^	>200 μM			NA	
Baicalin	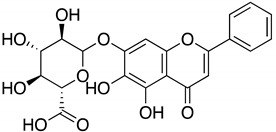		6.41 µM ^A^ 27.87 µM ^B^	>200 μM			NA	
Liu Shen	NA	MT123290.1	0.6024 μg/mL ^C^	4.930 μg/mL	Post-infection	Targeted both the virus and host factor. Down-regulate NF-κB/MAPK signaling pathway and reduce the production of proinflammatory cytokines.	NA	[55]
Lianhuaqingwen (LHQW)	NA	Genebank accession no. MT123290.1	411.2 μg/mL ^C^	1157 μg/mL	Immune modulation Virucidal	Reduced production of proinflammatory cytokines such as TNF-α, IL-6, CCL-2/MCP-1, and CXCL-10/IP-10. Altered the morphology of extracellular virus.	NCT05366231 (Completed, Phase 4)	[56]
Jinhua Qinggan (JHQG)	NA	NA	NA	NA	Immune modulation	Reduced production of IL-6 and increased the production of IFN-γ	NCT04723524 (Completed, Phase 2) NCT05507489 (recruiting, unknown phase)	[57]
Xuebijing (XBJ)	NA	NA	NA	NA	Immune modulation	NA	NA	[58]
NRICM101	NA	TCDC#4 from Taiwan CDC	0.22 mg/mL ^A^ 0.41 mg/mL ^F^ 0.28 mg/mL ^G^ 0.42–1.18 mg/mL ^H^	1.77 mg/mL	Pre- and post-infection Immune modulation	Blocked the binding of S protein to ACE2 Inhibited 3CL protease Reduced production of IL-6 and increased the production of TNF-α	NCT04664049 (Unknown status, unknown phase)	[59]
Mentha haplocalyx extract	NA	hCoV-19/Taiwan/4/2020	NA	NA	Prophylactic effect	NA	NA	[60]
		Natural extracts and their active ingredients						
Punicalagin (PUG) from Pomegranate peel extract (PPE)	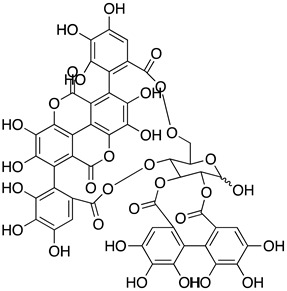	Isolate USA-WA1/2020	7.20 µM ^D^ 4.62 µM ^A^	100 µM	Polypeptide processing	Acted as an allosteric Inhibitor and inhibited catalytic activity 3CL^pro^.	NA	[61]
Artemisia annua L. extracts	NA	USA/WA12020	0.01–0.14 μg/mL ^C^	>500 µg/mL	Replication	NA	NA	[62]
EGYVIR	NA	hCoV-19/Egypt/NRC-03/2020	0.57 μg/mL ^D^	NA	Virucidal effect	Inactivated extracellular SARS-CoV-2. Modulated the immune system by inhibiting nuclear translocation of p50 and down-regulating Ikβα, TNF-α, and IL-6, thus preventing cytokine storm.	NA	[63]
*Netrium oleander* extract and oleandrin	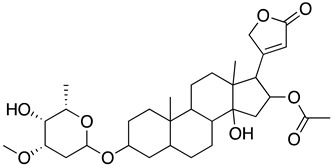	USA-WA1/2020 strain	7.07–11.98 ng/ml ^D^	>1 µg/mL	Prophylactic effect	NA	NA	[64]
*Vitis Vinifera* leaf Extract	NA	clinical isolate from Lazzaro Spallanzani Hospital	5–10 μg/mL ^D^	>500 μg/mL	Attachment and entry	NA	NA	[65]
Andrographolide from *Andrographis paniculata* extract	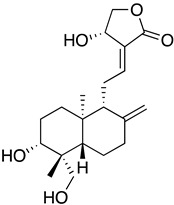	SARS-CoV2/01/human/Jan2020/Thailand	0.034 μM ^D^ 5–15.05 μM ^A^	13.19–81.52 μM	Polypeptide processing	Covalently linked to the active site of M^pro^.	NCT05019326 (Recruiting, unknown phase)	[66,67,68]
Hydroxytyrosol-Rich Olive Pulp Extract (HIDROX)	NA	JPN/TY/WK-521 strain	NA	NA	Virucidal effect	Interacted, changed the structure and aggregated the S protein.	NA	[69]
APRG64, mixture of *Agrimonia pilosa* (AP) and *Galla rhois* (RG)	NA	NCCP43326	NA	NA	Entry	Active ingredients such as ursolic acid, quercetin, and 1,2,3,4,6-penta-O-galloyl-β-d-glucose interacted with RBD of S protein.	NA	[70]
epigallocatechin gallate (EGCG) from green tea extract	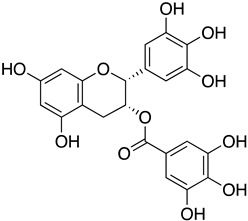	MUC-IMB-1	2.47 μg/mL ^E^	>20 μg/mL	Entry	Bound to RBD of S protein	NCT04446065 (Unknow status, Phase 2 and 3)	[71]
Propolis extract	NA	NA	NA	NA	Entry Replication	NA	NCT04680819 (Active unknown phase) NCT04480593 (Completed Phase 2 and 3) NCT04800224 (Completed Phase 2 and 3) NCT04916821 (Active, unknown phase)	[72,73]
Echinaforce	NA	BetaCoV/France/IDF0372/2020	NA	>100 µg/ml	Virucidal effect	NA	NCT04999098 (Recruiting, Phase 4)	[74]
Prunella vilgaris (NhPV) extract	NA	hCoV-19/Canada/ON-VIDO-01/2020, GISAID accession# EPI_ISL_425177	30 μg/mL ^E^	>200 μg/mL	Entry	Bound to ACE2 receptor and prevent viral entry	NA	[75]
		Pure natural compounds isolated from natural origin						
Isorhamnetin	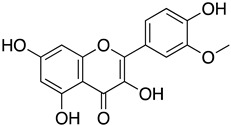	SARS-CoV-2 pseudovirus	NA	NA	Entry and polypeptide processing	Interacted with ACE2, S protein and inhibited the catalytic activity of M^pro^.	NA	[76,77,78]
Bufalin	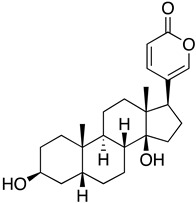	WIV-04	0.018 μM ^C^	>2000 μM	Replication	Targeted the ion exchange function of Na^+^/K^+^-ATPase	NA	[79]
Naringenin	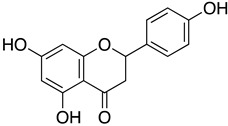	hCoV-19/Egypt/NRC-03/2020 (Accession Number on GSAID: EPI_ISL_430820)	28.35 µg/mL ^C^ 92 nM ^A^	178.75 µg/mL	Polypeptide processing	Inhibited catalytic activity M^pro^.	NA	[80]
Resveratrol	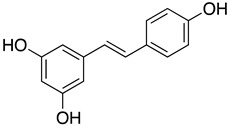	NL/2020 (EVAg-010V-03903)	66 µM ^D^	NA	Post-entry	NA	NCT04666753 (Completed, unknown phase) NCT04799743 (Recruiting, unknown phase)	[81]
Pterostilbene	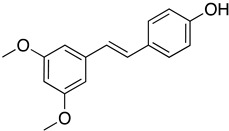		19 µM ^D^				NA	
Sulfated polysaccharide (RPI-27)	NA	NA	83 nM ^D^	>500 µg/mL	Entry	Interacted with RBD on S protein.	NCT04777981 (Not yet recruiting, unknown phase)	[82]
Crude polysaccharide 375	NA	WIV04	0.48 µM ^A^ 27 nM ^B^	136 mg/Kg on mice	Polypeptide processing	Inhibited catalytic activity M^pro^.		[83]
Sea cucumber sulfated polysaccharide	NA	NA	9.10 μg/mL ^D^	NA	Entry	Interacted with S protein.		[84]
Sinapic acid	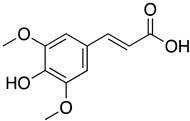	NA	2.69 µg/mL ^C^	189.3 µg/mL	Entry Assembly and release	Interacted with E protein	NA	[85]
X-206	NA	NA	14 nM ^C^	8.2 µM	Pre- and post-infection stage	NA	NA	[86]
Niclosamide	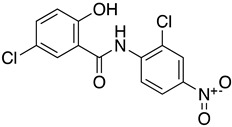	England/IC19/2020 (IC19)	0.34 µM ^D^	NA	Replication	Blocked intracellular calcium release and prevented syncytia formation	NCT04399356 (Completed, phase 2) NCT05087381 (Completed, phase 4)	[87]
lycorine	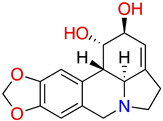	WIV04	0.31 µM ^B^	>40 μM	Post-infection	NA	NA	[88]

IC_50_ = the inhibitory concentration of natural compounds required to cause 50% inhibition of virus. CC_50_ = the cytotoxic concentration of natural compounds required to cause 50% cell death. 3CL^pro^ = 3C-like protease; S = S protein; Na^+^/K^+^-ATPase = Sodium–potassium adenosine triphosphatase; RdRp = RNA-dependent RNA polymerase; M^pro^ = Main protease; RBS = Receptor-binding domain; E = Envelope protein; TNF-α = Tumor necrosis factor alpha; IL-6 = Interleukin 6; NF-κB/MAPK = Nuclear factor *kappa B/*mitogen-activated protein kinase; TPGS = D-α-tocopherol polyethylene glycol succinate. ^A^ IC_50_ value is measured in terms of viral protein inhibition. ^B^ IC_50_ value measured in terms of reduction in SARS-CoV-2 RNA copy number, measured using qRT-PCR. ^C^ IC_50_ value measured in terms of cytopathic effects induced by SARS-CoV-2. ^D^ IC_50_ value measured in terms of reduction in SARS-CoV-2 plaque or foci number. ^E^ IC_50_ value measured in terms of inhibition of entry of psuedotyped virus. ^F^ IC_50_ value measured in terms of binding of viral protein to host cell receptor. ^G^ IC_50_ value measured in terms of viral protein expression. ^H^ IC_50_ value measured in terms of cytokine reduction. ^1^ oral liquid form of Shuanghuanglian preparations against 3CL^pro^. ^2^ lyophilized powder of Shuanghuanglian preparations against 3CL^pro^. ^3^ oral liquid forms of Shuanghuanglian preparations against PL^pro^. ^4^ lyophilized powders of Shuanghuanglian preparations against PL^pro^.

## Data Availability

Not applicable.

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
