# Peer review of "COVID-19 Therapeutic Potential of Natural Products"

_ijms, 2023, doi:10.3390/ijms24119589_

Round 1

Reviewer 1 Report

In this review, the authors summarized current evidence on the use of natural products for COVID-19 prevention and/or treatment. The review is well written, the topic is interesting and up to date; however, many parts of the manuscript need to be improved. Here below, I report a list of my concerns:

1) Introduction: is well written but I feel that concepts are sometimes unbound and not well organized. I suggest the authors to improve its organization by a)reporting first the most important natural compounds for each class; b) reporting the comparisons between natural products and synthetic products. For more detail, see the comments below:

-line 27: update the sentence to nowadays (last access September 2021). 

-lines 30-37: add some references showing the impact of COVID-19 in low-income countries (the lines are not referenced). 

-line 45: write out the acronym nsps and correct it with NSPs.

-Line 107: the main classes of natural compounds should be listed before introducing each single class. Placing table 1 and figure 2 (describing classes in general) within the paragraph about pre-infection inhibitors makes the introduction disorganized and somewhat confusing. Please put the list of classes and reference to table 1 and figure 2 after line 107.  

-General consideration: to add novelty to the present review, compared to similar previous ones, I would add information on RCTs showing the role of natural products in humans with COVID-19. For instance, Nicolussi et al. (10.3390/microorganisms10020211) grouped clinical evidence of the role of echinacea in COVID-19; similarly, resveratrol showed some evidence of improved prognosis in COVID-19 patients (10.21203/rs.3.rs-861831/v1); at the end of the description of each compound, add some information on RCTs and clinical studies reported in literature (adding a further column reporting available clinical evidence to table 1 could increase the quality of the paper). Similarly, do for all the molecules and classes listed. 

-line 124: change "usually use" with "are usually used".

-Lines 126-134: the comparison between natural compounds and synthetic ones (e.g. hydroxicloroquine and umifenovir) should be placed after listing and briefly discussing natural compounds carrying pre-infection inhibition properties. Which are the main natural pre-infection inhibitors available today (as reported in Figure 2?). 

-line 130: "higher and longer doses" is not clear. Maybe you want to say "high doses and long regimens"

-line 136: include more references; you stated that "mounting evidence supported the use of natural compounds in COVID-19", but you cited only reference 40 which is  related to green tea and black tea polyphenols (which are not described). Please integrate the list of references with other studies. 

-Line 150: write out the acronym SPR.

-Line 153: add reference after "remdesivir".

2) Methods: a brief paragraph including search strategy should be added after stating the aim of the study. Furthermore, the aim of the study should also include a statement regarding the novelty of this review compared to previously written ones (e.g. including evidence from new RCTs).

English is fine and only minor checks are needed.

Author Response

Dear Reviewer,

Thank you for your time and your comments on our manuscript. Attached is our revised manuscript which is amended according to your comments and you can find a file with reply to your comments. 

Thank you

Reviewer 2 Report

I was really interested in reviewing this article considering the importance of finding effective but also accessible alternative therapeutic resources for COVID-19.

I consider that this paper is a very comprehensive review, with scientific and promising practical value. It is broad based, yet relatively detailed. The information is well organized and updated, based on a large number of well-chosen references. The iconography is representative and facilitates the understanding of the theoretical data.

I appreciate section 6 which emphasizes the promising idea of combining various types of therapy and discusses the hot topics for prospective research regarding the treatment for COVID-19.

The conclusions are also discussed in a proper manner.

All these aspects leads me to recommend the publication.

Author Response

(The authors gave the same response as above.)

Round 2

Reviewer 1 Report

My sincere congratulations to the authors of the manuscript. The version now provided has been thoroughly revised and is more clearly described in all its parts (both text, figures, tables and references). They have addressed all critical points, and now the manuscript can be accepted for publication.